# Glidescope Video Laryngoscopy in Patients with Severely Restricted Mouth Opening—A Pilot Study

**DOI:** 10.3390/jcm12155096

**Published:** 2023-08-03

**Authors:** Zohal Popal, André Dankert, Philip Hilz, Viktor Alexander Wünsch, Jörn Grensemann, Lili Plümer, Lars Nawrath, Linda Krause, Christian Zöllner, Martin Petzoldt

**Affiliations:** 1Department of Anesthesiology, Center of Anesthesiology and Intensive Care Medicine, University Medical Center Hamburg-Eppendorf, Martinistrasse 52, 20246 Hamburg, Germany; z.popal@uke.de (Z.P.); p.hilz@uke.de (P.H.); v.wuensch@uke.de (V.A.W.); l.pluemer@uke.de (L.P.); n.nawrath@uke.de (L.N.); c.zoellner@uke.de (C.Z.); m.petzoldt@uke.de (M.P.); 2Department of Intensive Care Medicine, Center of Anesthesiology and Intensive Care Medicine, University Medical Center Hamburg-Eppendorf, Martinistrasse 52, 20246 Hamburg, Germany; j.grensemann@uke.de; 3Institute of Medical Biometry and Epidemiology, University Medical Center Hamburg-Eppendorf, 20246 Hamburg, Germany; l.krause@uke.de

**Keywords:** airway management D058109, intubation, intratracheal D007442, laryngoscopy D007828, respiratory system D012137

## Abstract

Background: An inter-incisor gap <3 cm is considered critical for videolaryngoscopy. It is unknown if new generation GlideScope Spectrum™ videolaryngoscopes with low-profile hyperangulated blades might facilitate safe tracheal intubation in these patients. This prospective pilot study aims to evaluate feasibility and safety of GlideScope^TM^ videolaryngoscopes in severely restricted mouth opening. Methods: Feasibility study in 30 adults with inter-incisor gaps between 1.0 and 3.0 cm scheduled for ENT or maxillofacial surgery. Individuals at risk for aspiration or rapid desaturation were excluded. Results: The mean mouth opening was 2.2 ± 0.5 cm (range 1.1–3.0 cm). First attempt success rate was 90% and overall success was 100%. A glottis view grade 1 or 2a was achieved in all patients. Nasotracheal intubation was particularly difficult if Magill forceps were required (*n* = 4). Intubation time differed between orotracheal (*n* = 9; 33 (25; 39) s) and nasotracheal (*n* = 21; 55 (38; 94) s); *p* = 0.049 intubations. The airway operator’s subjective ratings on visual analogue scales (0–100) revealed that tube placement was more difficult in individuals with an inter-incisor gap <2.0 cm (*n* = 10; 35 (29; 54)) versus ≥2.0 cm (*n* = 20; 20 (10; 30)), *p* = 0.007, while quality of glottis exposure did not differ. Conclusions: Glidescope^TM^ videolaryngoscopy is feasible and safe in patients with severely restricted mouth opening if given limitations are respected.

## 1. Introduction

Despite emerging developments and innovations in the field of airway management, difficult tracheal intubation is still a challenging situation in anesthesia, emergency and intensive care medicine and a main cause for anesthesia-related adverse events [1,2,3]. Restricted mouth opening is an important risk factor and a possible exclusion criterion for tracheal intubation with conventional direct laryngoscopes [4,5,6].

Videolaryngoscopes use cameras embedded on the blade tip of a laryngoscope that display the camera view of the glottis on a screen (indirect laryngoscopy) and hereby facilitate tracheal tube placement under improved visual control [7,8]. Simulation studies indicated a benefit of videolaryngoscopy in many clinical settings [9,10]. Videolaryngoscopy has been clinically established for more than two decades [11,12,13,14]. A recent Cochrane analysis revealed that videolaryngoscopes improve glottis exposure and prevent failed intubation, hypoxemic events and accidental esophageal intubation [15]. Hence, universal first-line or first-intend videolaryngoscopy became popular in adults in many hospitals [16,17]. Routine use of videolaryngoscopy has been recommended whenever feasible [18], awake videolarygoscopy became an established technique to manage expected difficult intubation [19,20] and the first validated universal classification for videolaryngoscopy was recently introduced [7,8].

Videolaryngoscopy, however, might be particularly helpful in individuals with restricted mouth opening, although robust prospective clinical data are rare. On the other hand, severely restricted mouth opening may alter blade insertion, blade advancement and glottis visualization, as well as tracheal tube alignment and placement during videolaryngoscopic-guided tracheal intubation; hence, restricted mouth opening is considered an important risk factor for difficult or failed videolaryngoscopy and a possible limitation of the method [5,6,21,22,23,24]. 

An inter-incisor gap <3.0 cm has been proposed to be a critical threshold for videolaryngoscopy [21,22,23,24,25,26]. However, study findings are still inconclusive and reasonable lower limits for inter-incisor gaps for videolaryngoscopy have never been systematically evaluated in larger prospective trials. Notably, especially in patients with restricted mouth opening the experience of the airway operator has to be considered a relevant cofactor for successful tracheal intubation with videolaryngoscopy [27].

Further, the specific blade shape and profile of videolaryngoscopes have to be considered; they particularly differ between Macintosh type and hyperangulated blades, but also between manufacturers. Most manufacturers provide two different shaped videolaryngoscopy blades: Macintosh type blades with a small angle that still allows for a direct view on the glottis, and hyperangulated blades that allow for a better view beneath the epiglottis, requiring less lifting force [28], but disqualify for direct laryngoscopy. A recent meta-analysis suggested that the improved glottis exposure achieved with hyperangulated blades might not necessarily translate into faster tracheal intubation [29]; however, it remains unclear if the improved view might translate into better intubation in terms of first pass or overall success rates [29]. Due to their specific geometry and function, hyperangulated blades might be particularly helpful for tracheal intubation in individuals with restricted mouth opening; however, currently, robust data that support this assumption are still lacking.

Nasotracheal intubation is a very common practice in patients with severely restricted mouth opening, especially in those undergoing oral and maxillofacial surgery [30]; this constellation has most commonly be considered a traditional domain of awake bronchoscopic intubation [19], and the role and limitations of videolaryngoscopy remain unclear.

The GlideScope^TM^ was the first commercially available videolaryngoscope and was introduced by Dr. John Pacey in 2001. Cooper et al. reported their first clinical experience in 2003 [13]. The latest version is the GlideScope Spectrum™ single-use videolaryngoscopes, which use hyperangulated videolaryngoscopy blades with a low-profile design (diameter: height 11 mm for the LoPro S3 and 12 mm for the LoPro S4 blade, Figure 1). From the theoretical point of view, this signature blade angle as well as the low profile might contribute to an improved maneuverability and enhanced working space in individuals with restricted mouth opining. However, it is unknown if these favorable attributes might improve glottis exposure or enable better, faster or safer tracheal intubation in individuals with small inter-incisor gaps. Currently, there is a lack of robust data regarding feasibility and safety of videolaryngosopy in individuals with severely restricted mouth opening ≤3.0 cm.

The aim of this prospective observational pilot study was to assess feasibility and safety of videolaryngoscopic intubation with the GlideScope Spectrum™ in individuals with severely restricted mouth openings. A secondary aim of the study was to compare the success rates, intubation times and subjective ratings of the airway operators regarding the quality of glottis exposure and ease of tube placement between individuals with inter-incisor gaps of <2.0 cm and ≥2.0 cm and between orotracheal and nasotracheal intubations with the GlideScope Spectrum™.

## 2. Materials and Methods

This single-center prospective observational cohort study was conducted in accordance with the Declaration of Helsinki. The design and reporting were adapted to the Strengthening the Reporting of Observational Studies in Epidemiology (STROBE) guideline (Appendix A). The study was approved by the Institutional Review Board (Ethics Committee of the Medical Board of Hamburg, Germany) on 9 July 2019 (PV6094) and registered prior to patient enrollment on Clinical-Trials.gov (NCT04174833, first posted 2 November 2019, Principle Investigator Martin Petzoldt). Written informed consent was obtained from each participant.

### 2.1. Patient Allocation

Adult patients ≥18 years that presented at our Anesthesia Preassessment Clinic before elective ear, nose and throat (ENT) or oral and maxillofacial (OMF) surgery between 20 January and 17 November 2021 were assessed for eligibility. Only individuals with severely restricted mouth openings between 1.0 and 3.0 cm for any reason scheduled for elective ENT or OMF surgery under general anesthesia and planned orotracheal and nasotracheal intubations were considered for inclusion. Pregnant or breastfeeding women and patients with confirmed indications for awake bronchoscopic intubation, such as progressive pharyngolaryngeal tumors, abscesses or other obstructive or space-consuming lesions, as well as loose teeth, anticipated difficult facemask ventilation, risk for rapid desaturation or risk for pulmonary aspiration, who qualified for rapid sequence induction, were excluded.

All patients underwent a structured preoperative physical examination and risk evaluation in accordance with standards laid out by the Department of Anesthesiology that comprises clinical history and physical examinations inclusive the upper lip bite test, the simplified airway risk index (SARI that incorporates the risk predictors: mouth opening, thyromental distance, Mallampati score, movement of the neck, underbite, body weight and history of previous intubations), the Wilson score (that incorporates weight, cervical spine and jaw mobility; degree of retrognathia; prominent incisors; and the inter-incisor distance) [4,6] and flexible nasendoscopy [31,32] if appropriate. Patients were systematically checked for indicators for awake tracheal intubation taking into account predictors for difficult tracheal intubation, suspected difficult facemask and/or supraglottic-airway ventilation, apnea intolerance and risk for aspiration [33].

### 2.2. Data Collection

All tracheal intubations were performed in the operation theater using GlideScope Spectrum™ single-use videolaryngoscopes (Verathon Inc., Bothell, WA, USA) with either LoPro S3 or LoPro S4 blades. GlideRite^®^ Stylets (Verathon Inc., Bothell, WA, USA) were used for all orotracheal intubations, while nasotracheal intubations were performed without a stylet. Indirect epiglottis lifting facilitated by point pressure on the hyoepiglottic ligament with the blade tip placed in the epiglottic vallecula was attempted first-line in all patients [34]. Either endotracheal tubes (Rüschelit^TM^, Teleflex Medical, Athelon, Ireland), cuffed reinforced endotracheal tubes (Woodbridge type; Mallinckrodt Lo-Contour^TM^, Covidien, Dublin, Ireland) or Shiley™ oral or nasal RAE tubes (RAE TrachealTubes with TaperGuard^TM^ Cuff; Covidien, Dublin, Ireland) were used for tracheal intubation. Highdosage rocuronium bromide was used to facilitate tracheal intubation, and adequate neuromuscular blockade was verified with train-of-four measurements (ToFscan^TM^, Dräger, Lübeck, Germany) in all patients.

Anesthesia induction; the choice of drugs, patient positioning and tracheal intubation; the choice of the blade size (either LoPro S3 or S4); and the use of airway adjuncts, tracheal introducer catheters, Magill forceps, airway optimization maneuvers and conversion to different intubation techniques and devices were left at the discretion of the anesthetists.

All participating physicians were consultant anesthetists experienced in the management of difficult airways. Furthermore, all physicians attended an at least 30 min structured manikin airway training inspired by the ‘Bath tea trolley training’ concept [35]. The years of physicians’ work experience were assessed within a questionnaire.

Study outcome variables, such as intubation time, intubation and laryngoscopy attempt or airway-related adverse events, were assessed by an independent study observer, while intubation-related variables such as increased lifting force were assessed by the airway operator. Further the airway operators subjectively rated the quality of glottis exposure, the ease of tube placement and the overall difficulty of videolaryngoscopic intubation on a visual analogue scale (0–100; lower values better).

All laryngoscopy videos were captured and reviewed by the airway operator and two additional independent raters (AD, PH) who assessed the percentage of glottis opening (POGO) [36] and videolaryngoscopic glottis view grades (six grades as proposed by Petzoldt [7,8] and coworker; modified after [32,33,34]): grade 1: vocal cords completely visible; grade 2a: part of the cords visible; grade 2b: posterior cords only just visible; 2c: arytenoids but not cords visible; grade 3: epiglottis but no glottis visible; and grade 4: laryngeal structures not visible). All raters were blinded to the ratings of each other. Discrepancies were discussed thereafter, and a consensus vote was reached in each case.

The inter-incisor gap, defined as the distance between patient’s upper and lower incisors with maximal mouth opening, was measured using a single-use measuring tape with an exact millimeter scale in the midline from the upper to lower teeth or gum before anesthesia induction (active mouth opening of the patient) and after anesthesia induction with complete neuromuscular blockade (passive mouth opening by the airway operator).

### 2.3. Sample Size

This is a pilot study with a hypothesis-generating, explorative character. The primary aim of the study was to prove feasibility and safety of GlideScope Spectrum^TM^ videolaryngoscopic intubation in individuals with severely restricted mouth opening. We considered that a case sample of 30 would be appropriate to first demonstrate feasibility.

### 2.4. Primary and Secondary Outcome Measures

The primary outcome measure to demonstrate feasibility was the overall success rate of videolaryngoscopic intubation with the GlideScope Spectrum^TM^, regardless of the number of attempts and time needed. Secondary outcome measures were the first attempt success (only one laryngoscopy and intubation attempt), intubation time (from the moment the device first touches the patient’s mouth until inflation of the tube cuff in the trachea), time to the best view gathered by videolaryngoscopy, percentage of glottis opening (POGO) [36], videolaryngoscopic glottis view grades (six grades [7,8]), the videolaryngoscopic intubation and difficult airway classification (VIDIAC) score [7,8], difficult facemask ventilation, airway-related adverse events as previously defined [2], recommendation for awake tracheal intubation recorded by the airway operator on an airway alert card (for future anesthetics), subjective rating of the quality of glottis view, the ease of tube placement and the overall difficulty of videolaryngoscopic intubation (visual analogue scales 0–100, lower values better).

### 2.5. Descriptive Statistics

Sample characteristics are given as absolute and relative frequencies or mean (standard deviation) as well as median (interquartile range), whichever is appropriate. Differences between orotracheal and nasotracheal intubations as well as between inter-incisor gaps <2.0 cm and ≥2.0 cm were compared using the Student’s *t*-test. A two-tailed *p* < 0.05 was considered statistically significant. We report nominal *p*-values without correction for multiplicity. Statistical analyses were performed using IBM SPSS Statistics version 29.0.0.0 (IBM, Armonk, NY, USA).

## 3. Results

Within the study period between 20 January and 17 November 2021, 2268 adults scheduled for elective tracheal intubation for ENT or OMF surgery were assessed for eligibility and 31 patients with severely restricted mouth opening who fulfilled all eligibility criteria were included (Figure 2). One of the patients dropped out because surgery was cancelled. The dataset of this analysis is complete without missing values.

Baseline characteristics of the study cohort are given in Table 1. Ten skilled anesthetists with a mean (SD) professional work experience of 16.3 (±6.3) years participated after an extended, focused mannequin pretraining. In total, 9 patients (30%) underwent orotracheal intubations and 21 (70%) underwent nasotracheal intubations with the GlideScope Spectrum™. The mean inter-incisor distance before anesthesia induction was 2.2 ± 0.5 cm and ranged between 1.1 and3.0 cm. Overall, 20 participants had an inter-incisor gap ≥2.0 cm and 10 had an inter-incisor gap <2.0 cm. The mean inter-incisor gap increased after anesthesia induction and neuromuscular blockade (2.3 ± 0.5 cm). However, in three individuals, the inter-incisor gap decreased after anesthesia induction. The main reasons for the restricted mouth openings were pain, temporomandibular joint dysfunctions, jaw fractures, cervicofacial flaps for head and neck reconstruction and/or tumors.

Tracheal intubation with the GlideScope Spectrum™ was successful in all participants. The first attempt success rate was 90%. Vocal cords were either completely (glottis view grade 1) or partly (glottis view grade 2a) visible [7,8] with the GlideScope Spectrum™. Videolaryngoscopic intubation was severe in 1 patient (VIDIAC score ≥ 3), hard in 2 patients (VIDIAC score 2) and easy or moderate (VIDIAC score ≤ 1) in the other 27 patients [7,8]. The glottis view grades were significantly better during nasotracheal intubation compared with orotracheal intubation (*p* = 0.048) and in individuals with an inter-incisor gap ≥2.0 cm compared with those with an inter-incisor gap <2.0 cm (*p* = 0.019). Further the POGO score was significantly higher in individuals that underwent nasotracheal intubation compared with those that underwent orotracheal intubation (88.2% versus 77.0%; *p* = 0.012) (Table 2).

The intubation time significantly differed between orotracheal (33 (25; 39) s) and nasotracheal (55 (39; 94) s); *p* = 0.049 intubations. We did not find a difference in intubation times between individuals with inter-incisor gaps ≥2.0 cm (36 (32; 62) s) and those with inter-incisor gaps <2.0 cm (59 (44; 104) s); *p* = 0.163 (Figure 3). Nasal intubations were reported to be particularly difficult if Magill forceps were required (*n* = 4). Figure 4 illustrates that time differences between individuals with an inter-incisor gap ≥2.0 cm and <2.0 cm predominantly rely on two extreme cases with a time to intubation above 120 s (Table 2).

Airway operators subjective ratings on visual analogue scales (0–100; lower values better) revealed that tube placement was more difficult in individuals with an inter-incisor gap <2.0 cm (35 (29; 54)) versus ≥2 cm (20 (10; 30)), *p* = 0.007, while the quality of glottis view did not differ between inter-incisor gap <2.0 cm (15 (4; 36)) and ≥2.0 cm (10 (5; 18)), *p* = 0.206 (Table 2).

In nine patients (30%), the airway operators issued an airway alert card after videolaryngoscopic intubation in order to warn physicians for future anesthetics; however, only in a single patient with a mouth opening of only 1.1 cm, the airway operator recommended awake bronchoscopic intubation for future anesthetics (Table 2).

## 4. Discussion

In this prospective pilot study, we were able to demonstrate feasibility and safety of GlideScope Spectrum^TM^ videolaryngoscopy in 30 patients with severely restricted mouth opening (1.1 to 3.0 cm) undergoing ENT or OMF surgery. Videolaryngoscopy was successful in all 30 patients. Due to the underlying diseases and planned surgery, many of these patients required nasotracheal intubation. Although overall glottis exposure was good with the GildeScope Spectrum^TM^ in these patients, airway operators rated that tube placement was more difficult in individuals with inter-incisor gaps <2.0 cm and particularly if Magill forceps were required for nasotracheal intubation. In most of the patients, airway operators recommended to use asleep videolaryngoscopic intubation if the patient required another tracheal intubation in the future; only in a single patient with a mouth opening of only 1.1 cm did the airway operator recommended awake bronchoscopic intubation for future anesthetics.

During preoperative airway preassessment and preselection of eligible patients, it should be considered that an active mouth opening as performed by an awake patient does not necessarily equal a passive mouth opening by the airway operator after anesthesia induction. In our case series, the inter-incisor distance decreased in 10% of the patients after anesthesia induction and neuromuscular blockade. A previous retrospective study already reported difficulties achieving full mouth opening after anesthetic induction in 20% of the patients with oral cavity or oropharyngeal cancer, in whom adequate mouth opening was assessed preoperatively [41]. Pain, temporomandibular joint dysfunctions, jaw fractures, cervicofacial flaps for head and neck reconstruction and/or tumors were the main reasons for the restricted mouth openings in our study.

For orotracheal intubation, hyperangulated videolaryngoscopes are most commonly used in conjunction with the corresponding hyperangulated stylets that complement the angle of the blade and thus optimize the advancement of the tracheal tube through the laryngeal inlet guided by the videolaryngoscope camera. However, surgical patients with severely reduced mouth opening often require nasotracheal intubation (70% of the participants in our cohort). For nasotracheal intubation with hyperangulated videolaryngoscopes, stylets cannot be used; hence, maneuverability of the tube in the pharynx and advancement of the tube through the laryngeal inlet might be affected. In these cases, Magill forceps are typically used to improve tube advancement [42].

Our data illustrate that intubation time was significantly longer for nasotracheal intubation than for orotracheal intubation. However, the mean intubation time of 55 s for nasotracheal intubation in our study is similar to the one reported for nasotracheal intubation with the GlideScope Spectrum^TM^ in individuals without restricted mouth opening in previous studies [42]. Notably, nasal intubation was particularly difficult and prolonged if Magill forceps were required. This might be due to the fact that the handling of the Magill forceps through a small inter-incisor gap is particularly challenging.

To our opinion, some important limitations of videolaryngoscopy in patients with severely restricted mouth opening must be considered when the decision for asleep tracheal intubation with a hyperangulated videolaryngoscope or awake brochnochoscopic intubation is made. First of all, it has to be considered that supraglottic airway devices are likely to fail in individuals with severely restricted mouth opening and, thus, disqualify as a fallback plan if airway management turns out to be more difficult than expected. Secondly, in our opinion, individuals with severely restricted mouth opening and an additional risk for aspiration, rapid desaturation or difficult facemask ventilation or expected difficult intubation due to progressive obstructive or space-consuming tumors should not be considered for asleep videolaryngoscopy; hence, they were excluded in our study [43,44].

Currently, awake bronchoscopic intubation is regarded standard of care in individual with severely restricted mouth opening, although thresholds are not well-defined [19,33,43,45]. Most guidelines do not provide thresholds for critical mouth openings that could be used for decision making for videolaryngoscopy or awake bronchoscopic tracheal intubation [12,19,33,43]. The French guidelines for ‘difficult intubation and extubation in adult anesthesia’ recommends not to use videolaryngoscopes if patients mouth opening is <2.5 cm without providing clear evidence for this recommendation [45].

Few studies with heterogenous design and inconsistent findings have addressed the issue of small mouth opening and videolaryngoscopy [20,21,22,23,46,47,48]. Three previous studies found that a mouth opening <3.0 cm was not an independent predictor for difficult [48] or failed [22] intubation or for first-pass intubation success [23] with the Glidescope^TM^. However, in all of these studies, it remains unclear why a threshold of <3.0 cm was chosen. In contrast, in 2016, Aziz and coworkers found that a mouth opening <3.0 cm was an independent predictor for difficult videolaryngoscopy with hyperangulated blades [21]; however, in this study, patients with inter-incisor gaps ≤2.0 cm were excluded. De Jong and coworkers used an inter-incisor gap <2.2 cm as an exclusion criterion for a videolaryngoscopy implementation program [49]. In a study of Cook and coworkers, small mouth openings were categorized into >5.0 cm, 5.0–4.0 cm, 4.0–3.0 cm and 3.0–2.0 cm, representing escalating difficulty classes. They found that smaller mouth openings were significantly associated with difficult intubation with the channeled Airtraq^TM^ videolaryngoscope. However, individuals with a mouth opening <2.0 cm were excluded and managed with awake bronchoscopic intubation [24].

This study has some limitations. Our data represent a single-center experience in patients undergoing ENT or OMF surgery and caution should be taken with caution when extrapolating them to other institutions or different patient populations. All tracheal intubations were performed by very skilled, specially trained consultant anesthetists; hence, the finding should not be extrapolated to less experienced airway operators.

## 5. Conclusions

This study demonstrated that GlideScope Spectrum^TM^ videolaryngoscopy is feasible and safe in patients with severely restricted mouth opening if applied by experienced airway operators. Given limitations, such as expected difficult facemask ventilation, suspected risk for rapid desaturation or for pulmonary aspiration, must be respected. This study demonstrated that overall glottis exposure was good; however, especially in patients with mouth openings <2.0 cm and those requiring nasotracheal intubation by means of a Magill forceps, restricted tube placement has to be expected. Further controlled trial are required to assess efficiency of GlideScope Spetrum^TM^ videolaryngoscopy in individuals with severely restricted mouth opening.

## Figures and Tables

**Figure 1 jcm-12-05096-f001:**
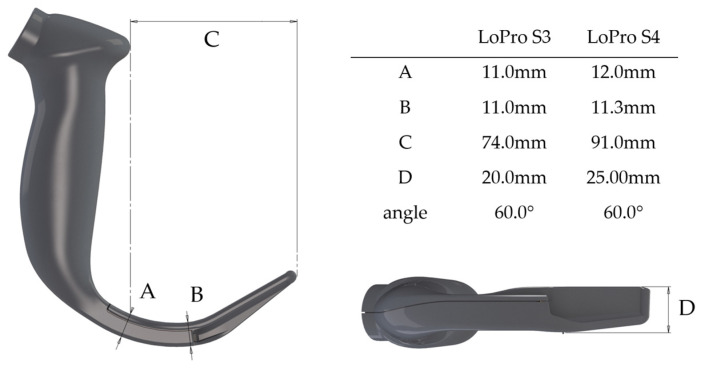
GlideScope^TM^ Spectrum LoPro blades, profile, angles and diameters (with permission from Verathon Inc., Bothell, WA, USA).

**Figure 2 jcm-12-05096-f002:**
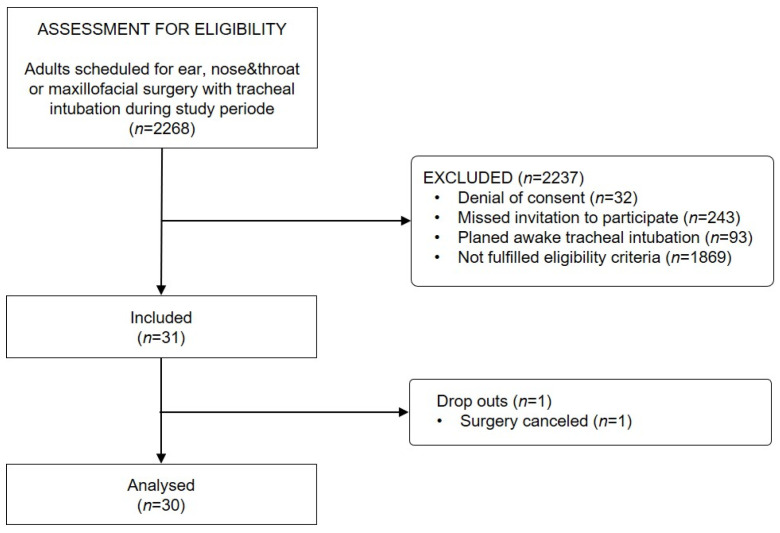
Study flow.

**Figure 3 jcm-12-05096-f003:**
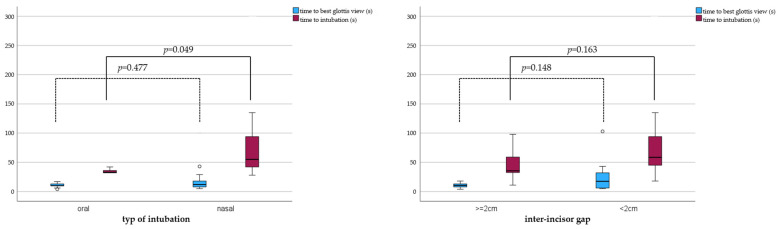
Boxplots showing pairwise comparisons between orotracheal and nasotracheal intubations with the GlideScope Spetrum™ (**left plot**) and between individuals with pre-induction inter-incisor gaps ≥2.0 cm and <2.0 cm (**right plot**) regarding the time to the best glottis view (blue boxes) and time to intubation (red boxes).

**Figure 4 jcm-12-05096-f004:**
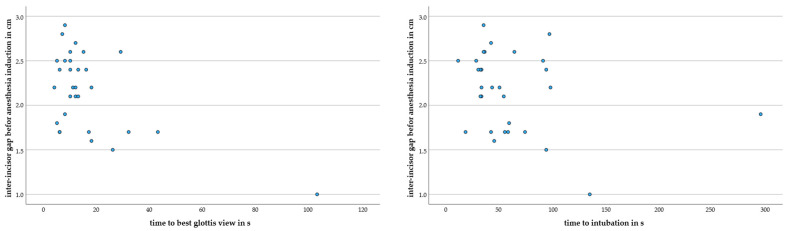
Scatter plots illustrating the relationship between the ‘time to best glottis view’ (**left**) or ‘intubation time’ (**right**) with the GlideScope Spectrum^TM^ on the *x*-axis and inter-incisor distance before anesthesia induction in cm on the *y*-axis.

**Table 1 jcm-12-05096-t001:** Baseline Characteristics.

Variables	(*n* = 30)
Sociodemographic data	
Age (years)	51.0 (27.8; 72.5)
Body mass index (kg m^−2^)	24.7 (20.7; 27.0)
Sex (male)	15 (50.0)
Preconditions	
ASA	
I	5 (16.7)
II	14 (46.7)
III	11 (36.7)
IV	0 (0.0)
History of previous difficult airway management	12 (40.0)
SARI score	5.17 ± 2.0
Wilson score	3.3 ± 1.8
Mallampati score	
Class I or II	0 (0.0)
Class III	11 (36.7)
Class IV	19 (63.3)
Neck mobility	
>90°	12 (40.0)
80–90°	13 (43.3)
<80°	5 (16.7)
Upper lip bite test	
Class I	3 (10.0)
Class II	7 (23.3)
Class III	20 (66.7)
Thyromental distance	
>6.5 cm	18 (60.0)
6–6.5 cm	4 (13.3)
<6.5 cm	8 (26.7)
Inter-incisor gap pre-induction *	2.2 ± 0.5
Inter-incisor gap post-induction *	2.3 ± 0.5
Retrognathia	6 (20.0)
Mandibula protrusion	6 (20.0)
Prominent incisor	5 (16.7)
Pharyngolaryngeal lesions	5 (16.7)
Dysphonia or dysphagia	2 (6.7)
Origin of restricted mouth opening (multiple choices possible)	
Pain	13 (43.3)
Inflammation	3 (10.0)
Jaw fractures	8 (26.7)
Temporomandibular joint dysfunction	5 (16.7)
Cervicofacial flap for head and neck reconstruction	5 (16.7)
Tumor	8 (26.7)
Craniofacial malformations	1 (3.3)

The dataset of this analysis is complete without missing values; values are mean ± SD or number (proportion), whichever is appropriate. * inter-incisor gaps were measured before anesthesia induction and after anesthesia induction with neuromuscular blockade. Abbreviations: ASA: American Society of Anesthesiology classification; SARI: simplified airway risk index.

**Table 2 jcm-12-05096-t002:** Outcome measures.

		Type of Intubations			Inter-Incisor Gap		
Variable	Overall	Oral *(n* = 9)	Nasal (*n* = 21)	*p*	≥2.0 cm (*n* = 20)	<2.0 cm (*n* = 10)	*p*
Success and attempts							
Overall successful intubation	30 (1.00)	9 (1.00)	21 (1.00)	N/A	20 (1.00)	10 (1.00)	N/A
First attempt success	27 (0.90)	9 (1.00)	18 (0.86)	0.083	19 (0.95)	8 (0.80)	0.314
Multiple laryngoscopy attempts	2 (0.07)	0 (0.00)	2 (0.10)	0.162	1 (0.05)	1 (0.10)	0.619
Multiple intubation attempts	1 (0.03)	0 (0.0)	1 (0.05)	0.522	0 (0.00)	1 (0.10)	0.343
Glottis exposure with videolaryngoscopy							
Glottis view grade ^#^				0.048			0.019
Vocal cords completely visible (grade 1)	15 (0.50)	2 (0.22)	13 (0.62)		13 (0.65)	2 (0.20)	
Part of the cords visible (2a)	15 (0.50)	7 (0.78)	8 (0.38)		7 (0.35)	8 (0.80)	
Posterior cords only just visible (2b)	0 (0.00)	0 (0.00)	0 (0.00)		0 (0.00)	0 (0.00)	
Cords not visible (2c or worse)	0 (0.00)	0 (0.00)	0 (0.00)		0 (0.00)	0 (0.00)	
POGO	86.0 (76.5; 95.5)	77.0 (50.0; 84.0)	88.2 (79.0; 97.5)	0.012	89.0 (78.5; 97.0)	77.5 (73.8; 88.5)	0.567
VIDIAC score				0.266			0.266
2	2 (0.07)	1 (0.11)	1 (0.05)		1 (0.05)	1 (0.10)	
≥3	1 (0.03)	1 (0.11)	0 (0.00)		0 (0.09)	1 (0.10)	
Time to best view [s]	12 (8; 17)	10 (8; 15)	12 (8; 18)	0.477	11 (8; 13)	18 (6; 35)	0.148
Intubation time [s]	44 (33; 78)	33 (25; 39)	55 (39; 94)	0.049	36 (32; 62)	59 (44; 104)	0.163
Difficult mask ventilation ^1^	8 (0.27)	1 (0.11)	7 (0.33)	0.161	4 (0.20)	4 (0.40)	0.258
Adjuncts and optimization maneuvers							
BURP	2 (0.07)	1 (0.11)	1 (0.05)	0.539	1 (0.05)	1 (0.10)	0.619
Magill forceps	4 (0.13)	0 (0.00)	4 (0.19)	0.042	3 (0.15)	1 (0.10)	0.716
Transition to a rescue technique	0 (0.00)	0 (0.00)	0 (0.00)	N/A	0 (0.00)	0 (0.00)	N/A
Airway-related adverse events ^2^	0 (0.00)	0 (0.00)	0 (0.00)	N/A	0 (0.00)	0 (0.00)	N/A
Recommendations for future anesthetics after tracheal intubation							
Awake bronchoscopic intubation recommended	1 (0.03)	0 (0.00)	1 (0.05)	0.522	0 (0.00)	1 (0.10)	0.343
Airway alert card issued	9 (0.30)	4 (0.44)	5 (0.24)	0.274	3 (0.15)	6 (0.60)	0.028
Subjective rating of the airway operator on visual analogue scale (0–100, lower values better)							
Quality of glottis exposure (0–100)	10.0 (5.0; 21.3)	25.0 (7.5; 27.5)	5.0 (3.0; 10.0)	0.080	10.0 (5.0; 17.5)	15.0 (3.80; 36.3)	0.206
Ease of tube placement (0–100)	27.5 (15.0; 40.0)	25.0 (17.5; 37.5)	30.0 (12.5; 40.0)	0.672	20.0 (10.0; 30.0)	35.0 (28.8; 53.8)	0.007
Overall difficulty of airway management (0–100)	20.0 (10.0; 31.3)	20.0 (15.0; 30.0)	15.0 (10.0; 35.0)	0.782	15.0 (10.0; 25.0)	25.0 (10.0; 62.5)	0.087

The dataset of this analysis is complete without missing values. Values are median (IQR) or number (proportion), whichever is appropriate. Reported *p*-values result from *t*-tests. ^1^ Difficult face mask ventilation was defined as Hun grad 2 or more [37]. ^2^ Airway-related adverse events are hypoxia, aspiration, swelling of the glottis, laryngospasm, dental or soft tissue injury and oral bleeding [2]; ^#^ grading of the glottis view gathered by videolaryngoscopy as proposed by Petzoldt and coworkers [7,8] (modified after [38,39,40]), Abbreviations: BURP: backward, upward and rightward pressure; POGO: percentage of glottis opening; VIDIAC: videolaryngoscopic intubation and difficult airway classification; VL: videolaryngoscopy.

## Data Availability

Not applicable.

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
