# Peer review of "Glidescope Video Laryngoscopy in Patients with Severely Restricted Mouth Opening—A Pilot Study"

_jcm, 2023, doi:10.3390/jcm12155096_

Round 1

Reviewer 1 Report

Dear authors 

thank you for preparing the present manuscript regarding difficult airway management. 

Although the topic is interesting I have a couple of concerns. 

In your methods you state a power calculation was not possible, but in your discussion you recite plenty of similar articles. Maybe you should have considered them as reference?

In addition to that the discussion is highly repetitive and not well written.

Please look at this part specifically, bring your results and reflect them to the other literature.

Some arguments are misleading and may need re-phrasing (see pdf).

The conclusion is too lengthy and again repeating the results.  

Dear editor, dear authors,

the English language use seems quite good.

However, a careful review preferably by a native speaker might improve the document.  

Author Response

REVIEWER #1

RECOMMENDATION 1: Dear authors, thank you for preparing the present manuscript regarding difficult airway management. Although the topic is interesting I have a couple of concerns. In your methods you state a power calculation was not possible, but in your discussion you recite plenty of similar articles. Maybe you should have considered them as reference?

RESPONSE: Thank you very much for pointing this out. We have to apologize, our statement was misleading. We cannot present a sample size calculation because the primary aim of the study is to demonstrate feasibility in 30 patients and not a comparison with a control group. Thus, there is no hypothesis or effect size where a primary hypothesis could rely on. This study is explorative and hypothesis generating. We now clearly stated this in the method section.

RECOMMENDATION 2: In addition to that the discussion is highly repetitive and not well written. Please look at this part specifically, bring your results and reflect them to the other literature. Some arguments are misleading and may need re-phrasing (see pdf). The conclusion is too lengthy and again repeating the results. 

RESPONSE: Thank you very much for your most helpful advise as well as your thoughtful and constructive criticism. Based on your recommendations we revised, restructured and streamlined the discussion section. We made all revisions that you suggested in the PDF, particularly:

  • Airway operator: We added in the methods section: ‘Notably, especially in patients with restricted mouth opening the experience of the airway-operator has to be considered a relevant cofactor for successful tracheal intubation with videolaryngoscopy.’, We deleted in the method section: ‘All physicians had performed at least 25 Macintosh and 25 hyperangulated video-laryngoscopic intubations.’ As this was misleading. All participating anesthetists were very skilled consultants [mean (SD) professional work experience of 16.3 (± 6.3) years] that have already performed some hundreds or thousands of videolaryngoscopic intubations.
  • Preoperative risk assessment (p 3, ln 129-136): We now present all assessed risk factors together with the scores in details.
  • Quality of muscle relaxation, we now state (p 4, ln 152): ‘High dosage rocuronium bromide was used to facilitate tracheal intubation and adequate neuromuscular blockade was verified with train-of-four measurements (ToF-scanTM, Dräger, Lübeck, Germany) in all patients.’
  • Three studies found that a mouth-opening <3 cm is not a predictor for difficult intubation, failed intubation or first pass success: This is correct; however it does not mean that a mouth-opening <3 cm is a predictor for easy intubation. It just outlines that the variable mouth opening <3 cm (y/n) is not able to discriminate between easy and difficult cases. Anyhow, studies are inconclusive, especially because the fourth study demonstrated that mouth opening <3 cm is a predictor for difficult videolaryngoscopy with hyperangulated blades. However, we understand your doubts, but we can only describe the current evidence and ambiguous study findings. To our opinion this clearly indicate that more studies in this field are needed.
  • Risk factor for difficult videolaryngoscopy in this context means that a difficult videolaryngoscopy is more likely to occur in individuals that present with a mouth opening < 3cm. As this finding is not coupled with an intervention, currently it is at the description of the anesthetist to decide if this patient should be managed by videolaryngoscopy or bronchoscopy or any other technique (it is simply a risk prediction model and not a decision-making aide).

Reviewer 2 Report

Except for 2 misspelings, everything else is acceptable. line 286: grades instead grads

line 378     opening instead oping

Acceptable, only 2 minor tipfalers

Author Response

REVIEWER #2

RECOMMENDATION: Except for 2 misspelings, everything else is acceptable. line 286: grades instead grads, line 378     opening instead oping; Comments on the Quality of English Language: Acceptable, only 2 minor tipfalers

RESPONSE: Dear reviewer #2, thank you very much for your interest in our scientific work, for reviewing our manuscript and for your great feedback. We revised these errors as recommended.

Reviewer 3 Report

I read with great interest the pilot study by Popal et al. on the video laryngoscopy in patients with severely restricted mouth opening. The manuscript is sound and well written. I have only minor comments to be addressed:

- Line 46. Besides the importance of videolaryngoscopy itself, authors should add that a growing body of literature has proven better performances of videolaryngoscopy compared to direct laryngoscopy in simulation studies, both for normal (doi: 10.26355/eurrev_202112_27620 - doi: 10.22514/sv.2023.053) and difficult airway intubation (doi: 10.1111/anae.14057 - doi: 10.22514/sv.2022.034). Please briefly discuss and add these 4 references.

- Line 82. Please first introduce the Glidescope as a relatively recent video laryngoscope (doi: 10.1007/BF03018651) and briefly discuss its characteristics, before introducing the GlideScope Spectrum.

- Line 132-134. Please specify all the test performed during preoperative evaluation. 

- Line 143-144. How was the blade chosen? Please specify.

- Line 156-157. Since 50 total intubations is a very small number, authors should not qualify consultants as expert in difficult airway management with this small cut off. If that was the case, I would suggest to separate data according to the experience. Otherwise, please correct the number.

- Please summarize the conclusions.

Author Response

REVIEWER #3

RECOMMENDATION 1: I read with great interest the pilot study by Popal et al. on the video laryngoscopy in patients with severely restricted mouth opening. The manuscript is sound and well written. I have only minor comments to be addressed:

- Line 46. Besides the importance of videolaryngoscopy itself, authors should add that a growing body of literature has proven better performances of videolaryngoscopy compared to direct laryngoscopy in simulation studies, both for normal (doi: 10.26355/eurrev_202112_27620 - doi: 10.22514/sv.2023.053) and difficult airway intubation (doi: 10.1111/anae.14057 - doi: 10.22514/sv.2022.034). Please briefly discuss and add these 4 references.

RESPONSE: Thank you very much for this good advice and for providing us with these interesting publications that we enjoyed reading. We now integrated these four cites in our manuscript as you recommended.

RECOMMENDATION 2: - Line 82. Please first introduce the Glidescope as a relatively recent video laryngoscope (doi: 10.1007/BF03018651) and briefly discuss its characteristics, before introducing the GlideScope Spectrum.

RESPONSE: Thank you very much for pointing this out. We now revised this section as recommended.

RECOMMENDATION 3: Line 132-134. Please specify all the test performed during preoperative evaluation.

RESPONSE: Thank you for this good advice. We now provide all details (p 3, ln 129): ‘All patients underwent a structured preoperative physical examination and risk evaluation in accordance with standards laid out by the Department of Anesthesiology that comprises clinical history and physical examinations inclusive the upper lip bite test, the simplified airway risk index (SARI that incorporates the risk predictors: mouth opening, thyromental distance, Mallampati score, movement of the neck, underbite, body weight, and history of previous intubations), the Wilson score (that incorporates weight, cervical spine and jaw mobility, degree of retrognathia, prominent incisors and the inter-incisor distance) and flexible nasendoscopy  if appropriate.‘

RECOMMENDATION 4: Line 143-144. How was the blade chosen? Please specify.

RESPONSE: We now state: (p 4, ln 155):

 (…) the choice of the blade size (either LoPro S3 or S4) (…) was left at the discretion of the anesthetists.

RECOMMENDATION 5: Line 156-157. Since 50 total intubations is a very small number, authors should not qualify consultants as expert in difficult airway management with this small cut off. If that was the case, I would suggest to separate data according to the experience. Otherwise, please correct the number.

- Please summarize the conclusions.

RESPONSE: We deleted the sentence: “All physicians had performed at least 25 Macintosh and 25 hyperangulated videolaryngoscopic intubations.” as this was only a formal prerequisite that we defined before conduction of our study. Finally only very skilled anesthetists with a mean (SD) professional work experience of 16.3 (± 6.3) years participated in our study; all of them have done several hundreds or thousands of videolaryngoscopic intubations.